# Biology and Therapeutic Properties of Mesenchymal Stem Cells in Leukemia

**DOI:** 10.3390/ijms25052527

**Published:** 2024-02-21

**Authors:** Cheng-Hsien Wu, Te-Fu Weng, Ju-Pi Li, Kang-Hsi Wu

**Affiliations:** 1School of Medicine, National Defense Medical Center, Taipei 114, Taiwan; samisconfident@gmail.com; 2Department of Pediatrics, Chung Shan Medical University Hospital, Taichung 402, Taiwan; mdsb1979@gmail.com; 3School of Medicine, Chung Shan Medical University, Taichung 402, Taiwan; 4Department of Pathology, School of Medicine, Chung Shan Medical University, Taichung 402, Taiwan

**Keywords:** mesenchymal stem cells, leukemia, hematopoietic stem cells, bone marrow microenvironment, clinical applications

## Abstract

This comprehensive review delves into the multifaceted roles of mesenchymal stem cells (MSCs) in leukemia, focusing on their interactions within the bone marrow microenvironment and their impact on leukemia pathogenesis, progression, and treatment resistance. MSCs, characterized by their ability to differentiate into various cell types and modulate the immune system, are integral to the BM niche, influencing hematopoietic stem cell maintenance and functionality. This review extensively explores the intricate relationship between MSCs and leukemic cells in acute myeloid leukemia, acute lymphoblastic leukemia, chronic myeloid leukemia, and chronic lymphocytic leukemia. This review also addresses the potential clinical applications of MSCs in leukemia treatment. MSCs’ role in hematopoietic stem cell transplantation, their antitumor effects, and strategies to disrupt chemo-resistance are discussed. Despite their therapeutic potential, the dual nature of MSCs in promoting and inhibiting tumor growth poses significant challenges. Further research is needed to understand MSCs’ biological mechanisms in hematologic malignancies and develop targeted therapeutic strategies. This in-depth exploration of MSCs in leukemia provides crucial insights for advancing treatment modalities and improving patient outcomes in hematologic malignancies.

## 1. Introduction of Mesenchymal Stem Cell (MSCs)

MSCs constitute a diverse subset of stromal stem cells obtainable from various adult tissues, demonstrating the unique capacity to differentiate into mesodermal lineage cells (adipocytes, osteocytes, and chondrocytes) and cells of alternative embryonic lineages [1]. Originally identified in bone marrow (BM), MSCs have since been isolated from diverse tissues without a universally accepted definition or quantitative assay for their identification within heterogeneous cell populations [2]. The International Society for Cellular Therapy has proposed minimum criteria for MSC definition, emphasizing plastic adherence; specific cell surface marker (CD73, CD90, and CD105) expression; the absence of hematopoietic and endothelial markers (CD11b, CD14, CD19, CD34, CD45, CD79a, and HLA-DR); and the capability to differentiate into adipocyte, chondrocyte, and osteoblast lineages in vitro [3]. While these criteria apply broadly, variations exist among MSCs derived from distinct tissue origins [4].

## 2. The Role of MSCs in BM Microenvironment

In the human BM niche, also known as the BM microenvironment, MSCs are crucial for regulating hematopoietic stem cells (HSCs) by providing physical support and secreting soluble factors. These interactions are vital for HSCs’ maintenance and fate, making MSCs integral to hematopoietic system development and differentiation [5,6,7]. MSCs and their surrounding microenvironment, comprising the extracellular matrix (ECM), neighboring cells, cytokines, hormones, and mechanical forces from the movement of the organism or the flow of physiological fluids, exhibit unique features but are interconnected both spatially and functionally [8]. MSCs influence their surrounding environment via various mechanisms, including immunomodulation, supporting hematopoiesis, and aiding in tissue repair [9]. Conversely, the microenvironment also plays a pivotal role in guiding the differentiation, growth, and functionality of MSCs [10]. Within a diseased microenvironment, MSCs can either exacerbate or mitigate the condition [11]. Therefore, the use of MSCs remains a subject of debate due to the paradoxical responses they may exhibit within the same microenvironment [12]. Understanding the intricate interplay between MSCs and their microenvironment is crucial for harnessing their therapeutic potential, especially in the context of diseases where the microenvironment may exert dual effects on MSC function and disease progression [9].

## 3. Leukemia Is a Cancer of HSCs

Leukemia, a diverse group of hematologic malignancies originating from BM, arises from the uncontrolled proliferation of developing leukocytes. Various genetic and environmental factors contribute to leukemia development, including ionizing radiation, benzene exposure, prior chemotherapy, viral infections, and genetic syndromes [13]. However, the etiology is not well-known. The BM niche is proposed to be associated with leukemia [14].

Pathophysiologically, leukemia originates from the malignant transformation of HSCs (Figure 1), leading to abnormal leukocyte production [15]. The progeny of these cancer cells has a growth advantage over normal cellular elements owing to an increased rate of proliferation, a decreased rate of spontaneous apoptosis, or both [16]. This result is a disruption of normal marrow function and, ultimately, marrow failure [17].

The classification includes acute or chronic types based on proliferation speed and myelocytic or lymphocytic types based on cell origin. Acute leukemia involves immature blasts, while chronic leukemia features partially mature cells accumulating in blood and organs, causing anemia and other complications. Subtypes encompass acute myeloid leukemia (AML), acute lymphoblastic leukemia (ALL), chronic myeloid leukemia (CML), and chronic lymphocytic leukemia (CLL) [18,19,20].

## 4. The Interactions of MSCs and HSCs in BM

The BM is a complex organ housing HSCs alongside cells derived from MSCs, like osteoblasts and adipocytes [21,22]. Together with the ECM, these cells create a specialized microenvironment crucial for mature hematopoietic cell formation and function [23]. MSCs play a pivotal role in supporting and regulating HSC properties [24]. Their interaction prevents HSC differentiation, protects against apoptosis, and promotes self-renewal [2,25].

Recent insights into the association of MSCs and HSCs in the BM have revealed a complex and dynamic niche. Early studies identified BM stromal progenitors expressing STRO-1 antigen and vascular-smooth-muscle lineage markers as contributors to hematopoiesis [26]. More recent findings have emphasized the significance of various MSC subsets, including CD146^+^ cells in the sinusoidal wall, CD271^+^ cells, CXCL12-abundant reticular cells, and Nestin^+^ MSCs, in establishing a specialized microenvironment for HSCs. These various subsets, each with specific functions, contribute to the intricate regulation of HSCs, emphasizing the dynamic and heterogeneous nature of the BM niche [27,28,29,30]. Notably, Nestin^+^ MSCs, particularly those located near sinusoids and associated with the sympathetic nervous system, play a crucial regulatory role by producing CXCL12. CXCL12, as well as stem cell factors, are essential soluble factors crucial for the maintenance of HSCs [30,31,32,33]. Future research should focus on further characterizing MSC subsets based on their function, physical localization, and cytokine secretion profile to achieve a comprehensive understanding and explore potential clinical applications [32,34].

On the other hand, there has been a longstanding suspicion that osteoblastic cells originating from MSCs play a crucial role in regulating primitive hematopoietic cells [35,36,37]. These cells produce hematopoietic cytokines and support B lymphopoiesis and myelopoiesis [38,39,40]. Studies manipulating osteoblast numbers revealed their critical role in HSC maintenance and self-renewal [41]. Adipocytes in the BM may inhibit hematopoiesis, limiting hematopoietic progenitor expansion while preserving the HSC pool [42].

MSC-derived extracellular vesicles (EVs) also play a crucial role in normal hematopoiesis [41]. EVs, categorized as exosomes and micro-vesicles, are membrane structures secreted by various cell types, including MSCs, T cells, B cells, and dendritic cells [43,44]. In the context of hematopoiesis, the hematopoietic niche relies on MSC-EVs, among other components, to maintain homeostasis and respond to stress or disease [45]. MSC-EVs contribute to the activation of HSCs in response to stimuli like hemorrhage, oxygen changes, chemotherapy, and irradiation [46]. These vesicles modulate the Wnt/β-catenin signaling pathway, enhancing HSC proliferation and inhibiting differentiation [24]. Additionally, MSC-EVs contain miRNAs that promote cell survival and proliferation and inhibit apoptosis or differentiation across various hematopoietic lineages [47]. Their immunomodulatory effects extend to macrophages, dendritic cells, neutrophils, NK cells, and T cells, influencing the overall balance in the immune system [48,49]. Moreover, MSC-EVs exhibit both pro-angiogenic and anti-angiogenic properties, contributing to angiogenesis under specific conditions [50,51].

## 5. MSCs in AML

AML represents approximately 80% of leukemia cases in adults [52]. Etiologically, AML has diverse risk factors, including myelodysplastic syndrome, congenital disorders like Down syndrome, environmental exposures (radiation, tobacco smoke, and benzene), and previous chemotherapeutic agent exposure [53,54,55,56]. AML’s pathophysiology involves mutations in hematopoiesis-related genes, resulting in clonal expansion of undifferentiated myeloid precursors [57,58]. AML is highly heterogeneous, categorized into favorable, intermediate, or adverse-risk groups based on cytogenetics [59]. Clinical and cytogenetic AML subgroups may show differences in MSC niches, highlighting the heterogeneity within AML [60,61].

### 5.1. MSCs in Pathogenesis and Progression of AML

MSCs play a crucial role in the BM microenvironment, influencing the pathogenesis and progression of AML. In AML, MSCs provide protection against apoptosis, and the bidirectional communication between leukemia cells and MSCs involves cell–cell interactions and soluble mediators [45,62]. Alterations in the marrow environment may contribute to leukemogenesis, and AML-MSCs exhibit distinct characteristics compared to their normal counterparts. AML-MSCs display enhanced adipogenic potential, as evidenced by increased oil Red O staining and surface expression of CD10 and CD92. Gene expression analysis reveals a model of adipogenic predisposition in AML-MSCs, with an underexpression of SOX9 and EGR2. SOX9 negatively influences adipogenic commitment by suppressing CEBP-α expression, while EGR2 acts as a positive regulator of adipogenesis. Adipogenic niches, reminiscent of those in solid tumors, promote tumor growth by inducing lipolysis and providing fatty acids to leukemia blasts, enhancing their survival. Targeting these adipogenic pathways in MSCs could disrupt the pro-tumoral niche, presenting a potential therapeutic strategy [63].

Exosomes derived from AML cells and AML-MSCs also play a role in altering the BM microenvironment [64]. They contain miRNAs that regulate gene expression in MSCs and contribute to leukemia cell survival [65]. In one study, it was shown that exosomes released by AML BM-MSCs delivered miR-10a to leukemia cells, leading to the downregulation of RPRD1A. This downregulation activated the Wnt/β-catenin pathway, providing protection to leukemia cells against the effects of chemotherapy [66]. Another study also showed that MSCs from AML contribute to the development and enhancement of drug resistance in AML cells. These AML-MSCs trigger changes in AML cells that resemble epithelial–mesenchymal transition (EMT), potentially linked to increased drug resistance. Moreover, this transformation is driven by the IL-6/JAK2/STAT3 pathway in AML, presenting a possible avenue for overcoming chemo-resistance in these cells [67]. Additionally, the immune-modulatory properties of AML-MSCs, demonstrated by their ability to induce regulatory T cells, suggest their involvement in shaping the immunosuppressive microenvironment supporting leukemic cells [68]. The interaction between AML blasts and MSCs results in the release of cytokines and chemokines, contributing to the formation of a leukemic niche [69].

### 5.2. Anti-Tumorigenic Effects of MSCs in AML

Controversial findings surround the effects of MSCs on cancer cells, with studies suggesting both inhibitory and proliferative activities [70,71]. The varied responses may be attributed to differences in experimental conditions, cell types, and signaling pathways [72]. In hematologic malignancies, including AML, MSCs exhibit the potential to either promote or inhibit tumor growth by modulating apoptosis or proliferation of tumor cells [62]. Studies indicate that MSCs interfere with hematologic malignancies by inducing cell-cycle arrest, inhibiting tumor growth, and suppressing the self-renewal ability of tumor cells. Notably, the mechanisms underlying these effects involve the secretion of cytokines, paracrine signals, and the modulation of various signaling pathways [73,74]. One study revealed EVs derived from BM-MSCs have in vitro inhibitory effects on leukemic cells, NB4 and K562 [75]. Additionally, MSCs have been explored for inhibiting vascular growth, inducing apoptosis of endothelial cells, and impairing angiogenesis in tumor environments [76].

Despite ongoing research, the mechanisms underlying the pro-tumorigenic and anti-tumorigenic effects of MSCs in AML remain incompletely understood. Further exploration of MSCs in hematologic malignancies is essential for unraveling their therapeutic potential and developing targeted interventions for AML treatment. The roles of MSCs in promoting and suppressing AML are summarized in Table 1.

## 6. MSCs in ALL

ALL is a hematologic malignancy characterized by the uncontrolled proliferation of abnormal B or T lymphoblasts, leading to BM and organ infiltration [77,78]. The etiology of ALL remains unknown, with possible association with genetic factors and/or environmental exposure having been implicated [79]. B-ALL is the most common malignancy of childhood [80]. The majority are under 18, with the peak age between two and ten years [81,82,83]. Pathophysiologically, ALL results from DNA damage causing the uncontrolled growth of lymphoid cells [84,85,86].

### 6.1. MSCs in B-ALL

#### 6.1.1. MSCs in Pathogenesis and Progression of B-ALL

In B-ALL, MSCs within the BM niche play a critical role in pathogenesis and progression. MSCs in B-ALL are known for their altered functionality and molecular profiles, particularly in their interactions with leukemic cells. These interactions involve various mechanisms, including the secretion of cytokines and chemokines, adhesion molecule interactions, and the transfer of metabolites and organelles [87,88].

A key aspect of MSCs in B-ALL is their ability to create a leukemia-supportive microenvironment. This includes the production of inflammatory mediators and transforming growth factor-β (TGF-β) family molecules, which can promote leukemic cell survival and progression [87]. MSCs also contribute to chemokine axis alterations within the BM niche, notably affecting the CXCL12/CXCR4 axis, which is crucial for leukemic cell homing and proliferation [89].

Another significant function of MSCs in B-ALL is their role in modulating the BM immune microenvironment. They can interact with various immune cells, potentially creating an immunosuppressive environment that favors leukemia progression [90]. Additionally, MSCs in B-ALL show changes in their ECM organization and osteoblastogenesis genes, indicating a disrupted BM niche conducive to leukemia advancement [91].

#### 6.1.2. MSCs in Chemo-Resistance of B-ALL

Recent studies reveal that MSCs contribute to chemo-resistance by providing amino acids and metabolites crucial for leukemic cell survival [92]. For instance, MSCs protect ALL cells from chemotherapy-induced amino acid depletion, particularly asparagine, via various mechanisms [93]. The exchange of EVs and tunneling nanotubes (TNTs) facilitates the transfer of essential components, including mitochondria, between MSCs and ALL cells, promoting chemo-protection [94]. Remarkably, shaking or using transwell inserts to break TNTs significantly reduces the survival of primary B-ALL blasts when treated with prednisolone, highlighting TNTs’ critical role in shielding leukemic cells from chemotherapeutic effects [94]. Additionally, recent studies show that TNTs enable the transfer of mitochondria from MSCs to B-ALL cells, enhancing leukemic cell resistance to ROS-inducing agents and chemotherapy, a mechanism confirmed in vivo in leukemia mouse models [95,96]. The adhesive interactions between MSCs and B-ALL cells are also crucial in promoting leukemia maintenance and chemo-protection. These interactions can activate various molecular pathways, like NF-kB and PI3K/Akt, enhancing leukemic cell survival and resistance to drugs [97,98,99]. MSCs’ release of matricellular proteins like periostin and osteopontin further facilitates these processes, creating a self-reinforcing loop that supports leukemia progression and chemo-resistance. Notably, targeting these ECM components presents a potential avenue for enhancing the efficacy of chemotherapy [100]. The pathogenesis, progression, and chemo-resistance of MSCs in B-ALL are summarized in Table 2.

### 6.2. MSCs in T-ALL

#### 6.2.1. MSCs in Progression of T-ALL

In T-ALL, MSCs have been shown to enhance the survival and proliferation of leukemia cells, contributing to T-ALL’s rapid progression. Notably, MSCs in the BM microenvironment have increased proliferation abilities and reduced apoptosis. The protective role of MSCs in T-ALL is mediated via direct contact and soluble factors, including cytokines and growth factors. In particular, the interaction between fibroblast growth factor 2 (FGF2) from MSCs and fibroblast growth factor receptor 2 (FGFR2) on T-ALL cells has been identified as a key mechanism driving T-ALL progression. These MSCs secrete FGF2, which interacts with FGFR2 on T-ALL cells, activating the PI3K/Akt/mTOR pathway and thus promoting T-ALL cell growth [102]. The use of Infigratinib, a tyrosine kinase inhibitor (TKI) targeting FGFR, has shown promise in inhibiting this interaction, suggesting a potential therapeutic approach for T-ALL [103,104,105]. This discovery highlights the significance of targeting the BM microenvironment, specifically MSCs and their interactions with leukemia cells, in developing effective treatments for T-ALL.

#### 6.2.2. The Role of MSCs in Chemo-Resistance of T-ALL Cells

In T-ALL, MSCs also play a complex role in treatment resistance. MSCs have been shown to provide protective effects to leukemia cells, shielding them from the impacts of chemotherapy [106]. This protection is primarily facilitated via two mechanisms: soluble factor-mediated and cell adhesion-mediated drug resistance [107]. Intriguingly, MSCs have been observed to transfer mitochondria to T-ALL cells, a process augmented by cell adhesion, which contributes significantly to chemo-resistance. This mitochondrial transfer not only reduces oxidative stress in leukemia cells by decreasing mitochondrial reactive oxygen species (ROS) levels but also aids in their proliferation and survival [108]. Furthermore, T-ALL cells treated with chemotherapeutic drugs like cytarabine or methotrexate can transfer mitochondria to MSCs, a novel discovery that adds to the understanding of mitochondria transfer in cellular interactions. This transfer, regulated by cell adhesion and mediated by structures like TNTs, plays a pivotal role in altering the redox balance within T-ALL cells, thereby inducing chemo-resistance. Additionally, the adhesion molecule ICAM-1 is particularly crucial in these mitochondria transfer processes [107].

Parthenolide (PTL), known to increase ROS stress and lower reduced glutathione (rGSH) levels, has shown promise as a therapeutic agent in T-ALL. It induces apoptosis by enhancing oxidative stress, but its efficacy may be compromised by MSCs, which can protect T-ALL cells from ROS stress and preserve rGSH levels. This protection is potentially via the release of cysteine, a key amino acid in rGSH production. Targeting the x_c_^−^ cysteine/glutamate antiporter system, which is involved in cysteine release and uptake, has emerged as a potential strategy to counteract the protective effects of MSCs. Inhibiting this system could potentially enhance the efficacy of drugs like PTL and overcome resistance mechanisms mediated by the leukemia microenvironment [109]. The progression and chemo-resistance of MSCs in T-ALL are summarized in Table 3.

## 7. MSCs in CML

CML is a myeloproliferative neoplasm characterized by the presence of the Philadelphia chromosome/translocation t(9;22)(q34;q11.2), resulting in the BCR-ABL1 fusion oncoprotein [110]. It predominantly involves proliferating granulocytes and affects both peripheral blood and BM [111,112]. The etiology of CML remains unclear. The pathophysiology of CML is defined by the BCR-ABL1 fusion oncoprotein. This oncoprotein acts as a constitutively expressed defective tyrosine kinase, affecting downstream pathways involved in cell growth, survival, apoptosis, and transcription factors [113,114]. Despite advancements in TKIs like imatinib, dasatinib, and nilotinib, CML treatment faces challenges due to the persistence of leukemia stem cells (LSCs) and incomplete molecular responses in many patients [115,116,117]. Resistance to TKIs, emerging at different stages of CML, poses a significant hurdle, underscoring the need for a deeper understanding of CML pathogenesis and more effective treatments. The role of MSCs in CML’s resistance to TKIs has garnered significant interest, particularly in the context of the BM microenvironment and its influence on disease progression. However, their specific contribution to the development and maintenance of CML is not fully understood.

### 7.1. MSCs in CML Pathogenesis and Progression

Studies have shown that MSCs in CML patients do not exhibit the BCR-ABL fusion gene, indicating they are not part of the leukemic process. This finding suggests that MSCs might retain normal functions despite the leukemic environment [118]. The functional behavior of MSCs is integral in either promoting or impeding LSCs’ expansion. Interestingly, MSCs from CML patients, despite their normal morphology and phenotype, appear to be functionally different from their healthy counterparts. Their ability to modulate the immune system is diminished. This impairment in CML-derived MSCs could contribute to the disruption of the normal hematopoietic environment [119].

### 7.2. The Roles of MSCs in Promoting Resistance to TKIs in CML

In CML, the primary therapeutic approach involves TKIs. These drugs target the BCR-ABL fusion protein but are not entirely effective in eliminating LSCs, leading to disease persistence or relapse. The resistance to TKIs in CML is multifaceted, involving BCR-ABL kinase mutations and the BM microenvironment’s role in providing survival signals to residual leukemic cells [120,121,122,123].

One of the critical factors in this environment is MSC-derived cytokines. Studies have identified IL-7, a hematopoietic cytokine produced by MSCs, as a significant player in conferring resistance to TKIs. It activates pathways like JAK1/STAT5, which helps CML cells survive even when BCR-ABL signaling is inhibited. This finding points to a complex relationship between MSCs and the leukemic process, where MSCs may indirectly contribute to drug resistance and disease progression. Therapeutic strategies in CML are thus evolving to include approaches that target the BM microenvironment, especially MSCs. The use of MSCs in combination with TKIs or other agents is being explored. For instance, manipulating the MSCs to alter their cytokine secretion or inhibit pathways like JAK1/STAT5 could enhance the effectiveness of existing treatments and potentially overcome TKI resistance [124].

### 7.3. The Multifaceted Impact of MSCs on CML

Recent studies have explored the potential of MSCs in altering the course of CML. Via their immunomodulatory and differentiation-inducing capabilities, these cells offer a promising avenue for CML treatment. One approach involves leveraging the ability of MSCs to restore impaired signaling pathways in CML cells. For instance, the MPL signaling pathway, crucial for megakaryocytic differentiation, is often disrupted in CML. MSCs can potentially rectify this by secreting thrombopoietin and activating downstream signaling pathways like JAK/STAT and p38 MAPK. This restoration could shift the balance from malignant proliferation to differentiation, thereby alleviating the disease [125].

Moreover, MSCs are being studied for their role in the immune response against CML. They have shown the ability to suppress lymphocytic proliferation in vitro, suggesting a potential role in modulating the immune system’s interaction with leukemia cells. This aspect is particularly important for understanding how the BM microenvironment, influenced by MSCs, can either support or hinder the proliferation of leukemic cells. The interaction between MSCs and the immune system in CML also extends to the influence of MSCs on myeloid-derived suppressor cells (MDSCs). MDSCs, which suppress immune responses and aid in tumor growth, are found in higher frequencies in CML patients. MSCs from these patients exhibit unique properties that could contribute to the generation and activation of MDSCs, thereby impacting the immune surveillance of leukemia [126]. The associations of MSCs in CML are summarized in Table 4.

## 8. MSCs in CLL

CLL is characterized by the proliferation and accumulation of mature but dysfunctional B lymphocytes [127,128]. CLL primarily affects the peripheral blood, spleen, lymph nodes, and BM [129]. The etiology of CLL remains uncertain, with possible association with genetic factors and/or environmental exposure [130,131,132,133,134,135,136]. The pathophysiology of CLL involves a two-step process leading to clonal B lymphocyte replication. Aberrant B-cell antigen receptor (BCR) expression induces cell-autonomous signaling, contributing to pathogenesis [137,138]. The disease progresses from monoclonal B cell lymphocytosis to CLL due to genetic abnormalities or changes in the BM microenvironment [139,140].

### 8.1. MSCs in CLL Progression

CLL involves complex interactions with the microenvironment, particularly with MSCs. This interaction, crucial for the survival and proliferation of CLL cells, occurs via various mechanisms, including direct cell-to-cell contact, soluble factors, and extracellular vesicles [141]. These interactions activate pathways like BCR and NF-κB, alter gene expression, and influence cell homing and survival [142]. Recent research has underscored the bidirectional nature of these interactions, where CLL cells can manipulate surrounding stromal cells, creating a supportive microenvironment. While ex vivo studies have provided insights into CLL/MSC interactions, there is a need for more in-depth in vivo analysis to fully grasp the dynamics of this relationship [141].

One study explored MSCs from the BM of 46 CLL patients, co-cultured to mimic in vivo conditions. It was found that MSCs support leukemic B cells’ survival more effectively than the stromal cell line HS-5. This co-culture of MSCs and CLL B cells exhibited varied survival rates, suggesting heterogeneity in leukemic clones’ interactions with their environment. Both soluble factors and cell-to-cell contact were found to support leukemic B cell viability, with certain cytokines and chemokines like IL-8, CCL4, CCL11, and CXCL10 being particularly influential [143,144]. The interaction between CLL B cells and MSCs is evident in another co-culture study, where MSCs have been found to protect CLL B cells from apoptosis and alter their expression of various markers. These interactions involve rapid activation of signaling pathways in MSCs, mediated by soluble factors released by CLL B cells. This bidirectional activation suggests a complex interplay between BM stromal cells and CLL B cells, influencing disease progression and treatment responses [145].

CLL cells have also been shown to regulate the transition of BM-MSCs to cancer-associated fibroblasts (CAFs) via exosomal miR-146a delivery. CLL cell-derived exosomes upregulate miR-146a in BM-MSCs, leading to the downregulation of ubiquitin-specific peptidase 16 (USP16) and an increase in CAF markers. This finding highlights the intricate ways in which CLL cells manipulate their microenvironment to support disease progression [146]. Furthermore, CLL cells’ secretion of platelet-derived growth factor (PDGF) modifies MSCs’ function, such as proliferation and migration, and enhances vascular endothelial growth factor (VEGF) production, influencing CLL progression. PDGF and VEGF levels in CLL patients correlate with aggressive disease features, suggesting their role in disease progression [147].

### 8.2. The Role of MSCs in Chemo-Resistance of CLL

MSCs were shown to protect leukemic B cells from apoptosis, even when exposed to chemotherapy agents like fludarabine and cyclophosphamide. This protection underscores the microenvironment’s role in CLL B cell survival and resistance to chemotherapy. The study also investigated the effects of kinase inhibitors like Bafetinib and Ibrutinib in CLL B cell–MSC co-cultures. These inhibitors induced apoptosis in leukemic cells, independent of MSC presence, by de-phosphorylating critical kinases in CLL pathogenesis. However, these treatments did not affect B cell migration towards an MSC-conditioned medium, suggesting that the cells retain their ability to migrate to protective niches [143]. The roles of MSCs in CLL are summarized in Table 5.

## 9. Potential Clinical Applications of MSCs in Leukemia

Stem cells, characterized by their dual capacities for differentiation and self-renewal, exist in two principal forms: embryonic stem cells (ESCs) derived from the blastocyst’s inner cell mass and adult stem cells with multipotency. While ESCs pose ethical concerns due to their association with tumorigenesis, adult stem cells offer a more ethically viable option for clinical applications, underscoring the attractiveness of adult stem cells for therapeutic use [148]. Accordingly, ongoing clinical trials seek to address concerns regarding the safety and efficacy of MSC therapy, emphasizing the importance of comprehending the dynamic relationship between MSCs and their microenvironment for successful therapeutic outcomes [9]. Renowned for self-renewal, pluripotency, and immunomodulatory potential, MSCs hold significant promise in cell therapy, tissue engineering, and regenerative medicine [149,150].

Cancer, characterized by uncontrolled cell growth and invasion, presents a challenge for targeted treatment due to its diverse nature. Promising therapies, including cell therapy and immunomodulation, are being explored, with MSCs emerging as potential candidates. Additionally, MSCs acting as delivery vehicles show promise in cancer cell therapy [151,152,153,154]. They possess hypo-immunogenic characteristics, migrate to tumor sites, and can be utilized for gene therapy [155]. Moreover, MSCs have been shown to suppress lymphocyte proliferation, offering potential therapeutic avenues, particularly in the context of GVHD following stem cell transplantation [156]. The mechanisms favoring clinical applications of MSCs in leukemia are presented in Figure 2. MSCs in leukemia are clearly different from healthy MSCs from BM. Modification of MSCs within the leukemic BM niche may be beneficial in the treatment of leukemia. However, there have been no studies using healthy MSCs to change the BM niche of leukemia until now, which is a good direction for further research.

### 9.1. The Role of MSCs in Hematopoietic Stem Cell Transplantation for Leukemia

In the field of leukemia treatment, the potential clinical applications of BM-MSCs are increasingly gaining attention. MSCs, known for their hematopoietic support capability, wide source availability, and low immunogenicity, have been primarily utilized in hematopoietic stem cell transplantation (HSCT) [157,158]. HSCT is one of the important treatments for leukemia. Their role, however, extends beyond mere support in HSCs’ recovery after HSCT.

### 9.2. The Antitumor Effect of MSCs in Hematologic Malignancies

Several mechanisms are proposed for the inhibition of hematologic malignancies by MSCs, including their potential use as delivery vehicles, inhibition of vascular growth, or induction of cell-cycle arrest [70]. MSCs, regardless of their tissue source and origin, have shown antitumor effects, suggesting their use in various cancer treatments [70,159]. The antitumor effects of MSCs are dependent on culture conditions, such as MSC concentration, which significantly affects proliferation rate, morphology, and secreted factors [160,161].

MSCs as delivery vehicles have gained interest, especially for their hypo-immunogenic characteristics and ability to migrate to tumor sites [154,155]. They can also serve as gene therapy carriers, delivering genes and other factors like IL-12, IL-24, and IFN-γ to tumor sites [155,162,163,164,165]. At the cellular level, MSCs exert effects via EVs containing miRNAs, RNA, and proteins that can be transferred to cancerous cells [166,167]. These EVs have demonstrated anti-proliferative effects on leukemic cells and cytotoxic effects in combination with chemotherapy drugs [168].

Regarding vascular growth inhibition, MSCs can impair vessel growth or angiogenesis under certain conditions, potentially important in hematologic cancers that depend on vascular support [169]. MSCs can migrate to endothelial cell-derived capillaries, producing ROS leading to apoptosis of endothelial cells and suppression of tumor growth [170,171].

The most common fundamental process in tumor growth inhibition by MSCs is cell-cycle arrest [73,172,173,174,175,176,177]. Although the mechanisms inducing cancer cell-cycle arrest by antitumor agents are yet to be fully identified, several studies have shown high levels of cells arrested at G0/G1, indicating MSCs’ role in inhibiting tumor proliferation [173,175,176,178].

### 9.3. Disrupt the Chemo-Resistance from MSCs in Leukemia

While MSCs are employed in therapies to enhance the efficacy of chemotherapy drugs, their interaction with LSCs via physical adhesion and cytokine–receptor interplay poses a challenge. Therapeutic strategies targeting MSCs in leukemia often focus on enhancing chemo-sensitivity and disrupting the protective microenvironment LSCs rely on. These strategies include the use of CXCL12/CXCR4 inhibitors, chemotherapy synergistic drugs, adhesion inhibitors, and bone homeostasis medicines [64]. CXCL12/CXCR4 inhibitors, such as plerixafor and other CXCR4 antagonists, have entered clinical trials, showing promise in disrupting the protective effects of MSCs on LSCs and increasing chemotherapy sensitivity [179,180]. Chemotherapy synergistic drugs are being investigated for their ability to jointly suppress the activity of LSCs and MSCs, particularly via targeting the WNT/β-catenin signaling pathway, essential for maintaining the stemness of both LSCs and MSCs [181,182,183]. These inhibitors aim to target myeloid leukemia cells and MSCs simultaneously for a synergistic effect [184]. Adhesion inhibitors, targeting molecules like CD44 and E-selectin, are being investigated to prevent LSCs homing and increase chemo-sensitivity by inhibiting the direct adhesion of LSCs to MSCs [185,186,187]. Furthermore, bone homeostasis medicines, like proteasome drugs carfilzomib and ixazomib, aim to remodel the leukemia BM microenvironment and induce apoptosis of leukemia cells [188,189].

## 10. Conclusions

The reported effects of MSCs on tumors are controversial, exhibiting both inhibitory and proliferative actions across various cancers [190]. In vitro, MSCs demonstrate tumoricidal effects on solid tumors such as breast and lung cancer while promoting proliferation in melanoma [191,192]. The in vivo context adds complexity, where MSCs inhibit pancreatic tumors but enhance growth in lung and prostate cancer models [193,194,195]. This dual role underscores the need to comprehend MSCs’ impact on tumor cell proliferation comprehensively. While understanding of MSCs in solid malignancies progresses, their role in leukemia remains less explored. Studies show that MSCs play a dual role in hematologic malignancies, with prevailing evidence suggesting they suppress both tumor cell proliferation and apoptosis [159,190,192], although some studies indicate a direct promotion of these processes [196,197]. Consequently, the therapeutic efficacy of MSCs in hematologic malignancies is uncertain, given the contradictory inhibitory and promoting effects observed in vitro and in vivo [151,172]. It is challenging to determine whether MSCs or HSCs are initially affected in leukemia. This is because when leukemia is diagnosed, both MSCs and HSCs within the BM usually have abnormalities, making it difficult to discern whether the HSCs or MSCs were initially affected.

The mechanisms governing the dual effects of MSCs in hematologic malignancies remain elusive. Some mechanisms favoring antitumor effects include MSCs serving as delivery vehicles, inhibiting vascular growth, and arresting the cell cycle [70]. However, these advantages are offset by MSCs’ role in aiding tumor blood vessel formation [198,199,200,201], influencing immune responses [202,203,204], and contributing to higher rates of metastasis and cancer relapse [205,206,207,208,209,210,211,212,213,214]. Furthermore, MSCs often shield tumor cells from the apoptosis triggered by drugs, contributing to increased resistance to chemotherapy [108,122,215,216,217,218,219,220]. As the understanding of MSCs’ intricate role in hematologic malignancies evolves, the challenge lies in delineating specific mechanisms for targeted therapeutic applications while addressing the complexities of their dual nature (Figure 3). Further research is imperative to unravel the full spectrum of MSC actions in hematologic malignancy progression, providing a foundation for developing effective and safe therapeutic strategies.

## Figures and Tables

**Figure 1 ijms-25-02527-f001:**
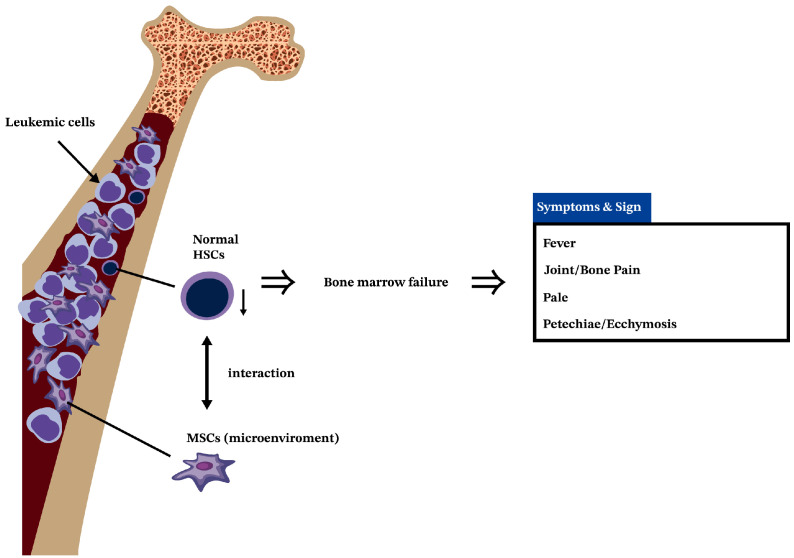
The pathophysiology of leukemia.

**Figure 2 ijms-25-02527-f002:**
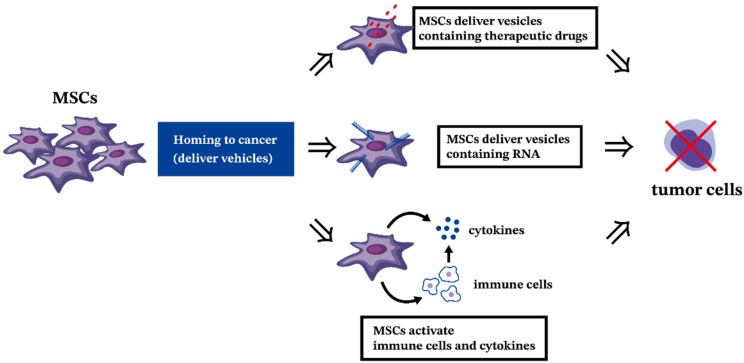
The potential clinical applications of MSCs in leukemia.

**Figure 3 ijms-25-02527-f003:**
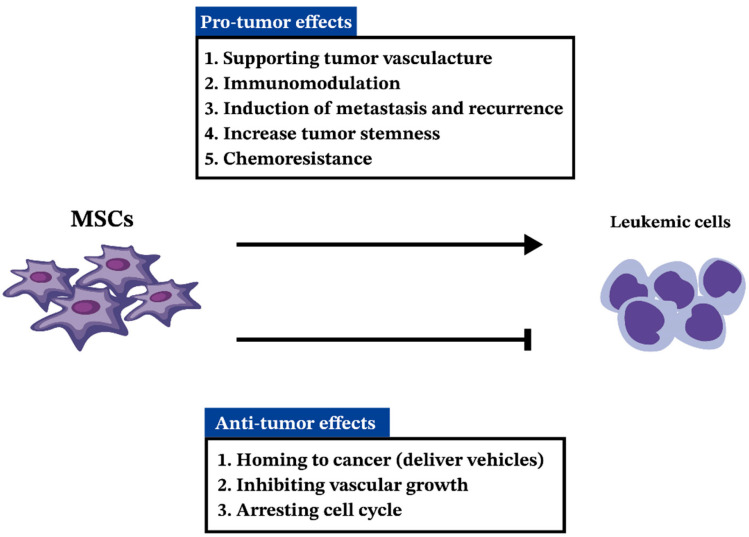
The pro-tumor effect and anti-tumor effect between MSCs and leukemic cells.

**Table 1 ijms-25-02527-t001:** The roles of MSCs in promoting and suppressing AML.

Reference	The Pro-Tumorigenic Effects of MSCs in AML
Brenner, A.K. et al. [62]	Even though the effects of single cytokines derived from BM-MSCs on human AML cells differ among patients, the final cytokine-mediated effects of the MSCs during co-culture are growth enhancement and inhibition of apoptosis.
Azadniv, M. et al. [63]	AML-MSCs possess adipogenic potential, which may enhance support of leukemia progenitor cells.
Barrera-Ramirez, J. et al. [65]	MSC-derived exosomal miRNA represents a potential mechanism for influencing gene regulatory networks in AML.
Wu, J. et al. [66]	AML BM-MSCs released exosomes that delivered miR-10a to leukemia cells and downregulated RPRD1A, which activated Wnt/β-catenin that subsequently conferred protection of leukemia cells from chemotherapy.
Lu, J. et al. [67]	AML-MSCs induce the EMT-like characteristics in AML; this phenotypic change could be related to chemo-resistance progression. Additionally, AML-MSCs induce the EMT-like program in AML via IL-6/JAK2/STAT3 signaling, which could be a target to reverse chemo-resistance in AML.
**Reference**	**The Anti-Tumorigenic Effects of MSCs in AML**
Liang et al. [73]	Direct contact with HFCL stromal cells could inhibit the proliferation and induce the differentiation of AML cells.
Phetfong, J. et al. [75]	This study with NB4 and K562 enlightens the influence of MSC-EVs on some types of leukemic cell lines. The molecular mechanism causing cell apoptosis seemed to be different between both cell lines.

**Table 2 ijms-25-02527-t002:** The pathogenesis, progression, and chemo-resistance of MSCs in B-ALL.

Reference	Main Findings
Fallati, A. et al. [87]	MSCs contribute to B-ALL pathogenesis and progression by creating a leukemia-supportive BM microenvironment rich in TGF-β molecules and inflammatory mediators.
Balandrán, J.C. et al. [101]	ALL tumor cells alter their environment, affecting MSCs and CXCL12 production. This change supports leukemic cell proliferation while hindering normal hematopoiesis.
Hughes, A.M. et al. [91]	BCR-ABL1^+^ B-ALL-associated MSCs exhibit reduced self-renewal capacity and extensive molecular alterations, indicating potential disruptions to important signaling pathways involved in inflammation, osteoblastogenesis, and ECM organization in vivo.
Chiu, M. et al. [93]	ALL blasts engage in an amino acid exchange with BM-MSCs, using glutamine to gain asparagine, aiding survival during L-asparaginase treatment. Blocking this pathway could enhance therapy effectiveness.
Polak, R. et al. [94]	TNT communication between B-ALL cells and MSCs in the BM influences cytokine release, impacting drug resistance and leukemia cell survival, suggesting TNT disruption as a potential therapeutic strategy.
Burt, R. et al. [96]	MSCs in ALL patient BM exhibit a cancer-associated fibroblast-like phenotype, secreting high levels of pro-inflammatory cytokines and affecting chemo-resistance, leading to a new hypothesis for preventing chemo-resistance in ALL.
Jacamo, R. et al. [99]	Leukemia cells modify BM-MSCs via NF-κB activation, enhancing chemo-resistance, suggesting that disrupting this interaction could improve leukemia treatment effectiveness.
Ma, Z. et al. [100]	Periostin, an extracellular component, enhances leukemia progression in B-ALL by increasing CCL2 expression via the integrin-ILK-NF-κB pathway, with reciprocal activation by CCL2, suggesting a key role in B-ALL cells-BM-MSC interactions.

**Table 3 ijms-25-02527-t003:** The progression and chemo-resistance of MSCs in T-ALL.

Reference	Main Findings
Tian, C. et al. [102]	BM microenvironmental MSCs promote T-ALL cell growth via FGF2 and FGFR2 interaction, activating the PI3K/AKT/mTOR pathway. Targeting this interaction with FGF2 inhibition or FGFR2 antagonism suppresses T-ALL progression.
Wang, J. et al. [107]	T-ALL cells combat chemotherapy-induced stress by transferring mitochondria to MSCs via TNTs, often adhering via ICAM-1, suggesting targeting mitochondria transfer as a strategy against T-ALL chemo-resistance.
Cai, J. et al. [108]	MSCs in direct co-culture with T-ALL cells enhance leukemia chemo-resistance by activating the MAPK/ERK pathway, leading to changes in mitochondrial dynamics and metabolism, suggesting targeting these interactions as a new treatment strategy.
Ede, B.C. et al. [109]	PTL, effective against T-ALL in xenografts, faces resistance, possibly due to the BM microenvironment. MSCs protect T-ALL cells by mitigating PTL-induced ROS stress, with the xc system playing a key role. Targeting this system may enhance PTL’s effectiveness against leukemia.

**Table 4 ijms-25-02527-t004:** The associations of MSCs in CML.

Reference	Main Findings
Jootar, S. et al. [118]	MSCs from CML patient BM, free of the Ph chromosome and capable of differentiating into osteoblasts, can support cord blood stem cell expansion and reduce graft-versus-host disease (GVHD) in stem cell transplants.
Xishan, Z. et al. [119]	MSCs from CML patients have impaired immunomodulatory functions, affecting the hematopoietic environment. This impairment might limit the effectiveness of autologous MSC transplantation in CML treatment, pointing towards allogeneic transplantation as a more viable option.
Zhang, B. et al. [121]	TKI treatment for CML is effective but does not eliminate LSCs, which can cause relapse. The BM microenvironment, particularly MSCs, protects these LSCs via N-cadherin and Wnt/β-catenin signaling, suggesting new treatment targets.
Vianello, F. et al. [122]	Human MSCs shield CML cells from imatinib-induced apoptosis; targeting the CXCL12/CXCR4 axis with anti-CXCR4 antagonists could enhance treatment efficacy.
Zhang, X. et al. [124]	IL-7, secreted by MSCs, plays a crucial role in CML by inducing resistance to TKIs via JAK1/STAT5 activation, suggesting a combined approach of IL-7/JAK1/STAT5 inhibitors with TKIs for effective treatment.
Zuo, S. et al. [125]	MSCs enhance megakaryocytic differentiation in CML by restoring MPL signaling, and when combined with MPL agonist Eltrombopag, significantly reduce leukemia burden, offering a new therapeutic strategy.
Giallongo, C. et al. [126]	In CML, MSCs play a crucial role in creating an immune-suppressive microenvironment by driving MDSC activation, affecting T lymphocyte function, and contributing to leukemia immune evasion, highlighting their potential as a therapeutic target.

**Table 5 ijms-25-02527-t005:** The roles of MSCs in CLL.

Reference	Main Findings
Binder, M. et al. [142]	BCRs on CLL cells recognize stromal cell antigens like vimentin, influencing apoptosis protection and potentially contributing to disease heterogeneity and progression.
Trimarco, V. et al. [143]	CLL B cell–MSC co-culture mimics BM conditions in vitro, reflecting CLL B cell response diversity. Kinase inhibitors like Bafetinib and Ibrutinib disrupt B cell–MSC interactions, crucial for targeting CLL treatment strategies.
Plander, M. et al. [144]	CLL cells alter BM-MSCs in the BM, enhancing their survival by upregulating specific adhesion molecules like ICAM-1, CD18, and CD49d and increasing secretion of cytokines like TNF-α, IL-6 and IL-8, which contribute to apoptosis resistance in CLL.
Ding, W. et al. [145]	In co-culture with MSCs, CLL B-cells show increased CD38, CD71, CD25, CD69, and CD70 markers, indicating disease progression. Soluble factors activate MSCs, leading to ERK and AKT phosphorylation and preventing both spontaneous and drug-induced apoptosis in CLL B-cells.
Yang, Y., Li, J. et al. [146]	In CLL, exosomes from CLL cells deliver miR-146a to BM-MSCs, inducing their transformation into CAFs by targeting USP16.
Ding, W. et al. [147]	The study demonstrates that PDGF secreted by CLL cells activates MSCs via the PDGFR-PI3K-Akt activation, increasing VEGF production and influencing leukemia progression, drug resistance, and angiogenesis in the CLL microenvironment.

## Data Availability

No new data were created or analyzed in this study.

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
