# Peer review of "Biology and Therapeutic Properties of Mesenchymal Stem Cells in Leukemia"

_ijms, 2024, doi:10.3390/ijms25052527_

Round 1

Reviewer 1 Report

Comments and Suggestions for Authors

Overall, a very good paper summarizing the MSC-HSC interactions in the BM niche. However, the "potential applications" section is the most important and requires significant development + critical analysis. 

MSCs in diseased tissue of leukemia are clearly very different from healthy MSCs obtained from the bone marrow or other sources. Are there any studies testing the utility of healthy MSCs within the cancerous BM-niche?

If so, could this be related to somehow the modification of MSCs within the bm niche? Also, how does the signature (i.e. immunomodulatory) differ between healthy and cancerous MSCs? 

Is there a way to test if early vs late onset leukemia affects MSCs first or HSCs first? Also, are there signals from HSCs are that are modifying the MSC immunophenotype? 

Reviewer 2 Report

Comments and Suggestions for Authors

Wu and others did a beautiful job in comprehensively reviewing the biology and therapeutic properties of MSC in leukemia. The review is well organized and presented. I enjoyed the tables summarizing the roles of MSC in multiple types of leukemia. The manuscript is already good for publication. I only have a minor suggestion to improve the manuscript further:

1. The anti and pro-tumorigenic effects of MSC in AML are quite interesting. Further investigations might be important to translate the MSC-targeting therapy to clinical settings. It would be helpful if the authors could improve the table 1. Instead of laying out the experimental results, it could be better to categorize the "pro-tumorigenesis" and "anti-tumorigenesis", and summarize the reasons why distinct results were derived from different experimental settings. 

2. Figure 2 and Figure 3 legends were swapped. I believe Figure 2 is the clinic application and Figure 3 is the anti/pro-tumorigenesis effects. 

Overall, it is a quality review that can be helpful for the scientists in the field. 
